# A Novel Dynamic Approach for Risk Analysis and Simulation Using Multi-Agents Model

Hassan Kanj [1,*], Wael Hosny Fouad Aly [1] and Sawsan Kanj [2]

1   College of Engineering and Technology, American University of the Middle East, Egaila 54200, Kuwait; wael.aly@aum.edu.kw
2   Data Scientist, Renault Group, 92100 Boulogne-Billancourt, France; sawsan.kanj@gmail.com
*   Correspondence: hassan.kanj@aum.edu.kw

**Abstract:** Static risk analysis techniques (SRATs) use event graphs and risk analysis assessment models. Those techniques are not time-based techniques and hence are inadequate to model dynamic stochastic systems. This paper proposes a novel dynamic approach to model such stochastic systems using Dynamic Fault Trees (DFT). The proposed model is called *Generic Dynamic Agent-Based Model* (GDABM) for risk analysis. GDABM is built on top of the well-known Agent-Based Modeling and Simulation (ABMS) technique. GDABM can model the dynamic system agents in both nominal (failure-free) and degraded (failure) modes. GDABM shows the propagation of failure between system elements and provides complete information about the system's configurations. In this paper, a complete detailed case study is provided to show the GDABM capabilities to model and study the risk analysis for such dynamic systems. In the case study, the GDABM models the risk analysis for a chemical reactor/operator and performs a complete risk analysis for the entire system. The GDABM managed to simulate the dynamic behavior of the system's components successfully using Repast Simphony 2.0. Detailed agent behavioral modes and failure modes are provided with various scenarios, including different time stamps. The proposed GDABM is compared to a reference model. The reference model is referred to as the ABM model. GDABM has given very promising results. A comparison study was performed on three performance measures. The performance measures used are (1) Accuracy, (2) response time, and (3) execution time. GDABM has outperformed the reference model by 15% in terms of accuracy and by 27% in terms of response time. GDABM incurs a slightly higher execution time (13%) when compared to the ABM reference model. It can be concluded that GDABM can deliver accepted performance in terms of accuracy and response time without incurring much processing overhead.

**Keywords:** multi agent system; failure analysis; dependent failures; risk analysis; agent-based simulation; stochastic systems; event-graphs; dynamic fault-trees

## 1. Introduction

The field of system engineering conducts risk analysis and assessment for various real-world industrial systems. Those systems have a high complexity level which depends on the huge size of the system, implying an important number of interactions between the system's components and its dynamic operational environment.

Analyzing risks related to such systems considers mainly two factors: the probability of having a failure and the severity of the resulting outcomes. This severity could vary from minor consequences to disastrous ones. Thus, modeling and simulation of such dynamic systems are crucial. Due to the complexity and the dynamic aspect of the studied systems, a dynamic representation of the system's behavior is needed to rank its performance and analyze its reliability.

In literature, modeling techniques are classified into static and dynamic. Static modeling techniques were used to assess system reliability, such as a bow-tie diagram, which

consists of an event tree and a fault tree, and block diagrams. Those techniques are not adequate for studying the dynamic effects of time-dependent systems; since there are many interesting behaviors that static modeling techniques will not be able to model. An example of such interesting behavior is time-dependent behavior that changes over time. On the other hand, and after many developments over the past decades, scientists developed many dynamic modeling techniques that provide means of modeling and simulation of the time-dependent behavior of various complex systems, including a wide range of real-world industrial systems operating in a dynamic environment. Those techniques overcome the limitations of the conventional static risk analysis techniques. Examples of such methods are Petrinets [1], graphical Markov models [2], state-transition graphs, Business Process Model [3], Stochastic Hybrid Automaton (SHA), Monte Carlo simulations [4], and Agent-Based Modelling and Simulation (ABMS) [5].

Yagi et al. [6] have proven that ABMS is one of the most adequate tools to model dynamic systems with autonomous agents. ABMS was proven to be suitable for risk assessment since the agents frequently cooperate and interact with each other [7]. To the best of our knowledge, ABMS is not widely used in the field of risk analysis and its application is limited by performing a general risk analysis without providing any details about the failure (causes and consequences), the failure propagation between the various system's components and the mutual relation between agents behavioral and failure modes.

This paper aims to (1) propose an extension to the classical ABM that overcomes the above limitations, (2) provide full details about risk analysis, (3) represent the failure propagation and risk analysis in the agent-based model, and (4) study the failure of a system' component to show its effects on the agent's behavioral mode in addition to its transition to the other system' components. This extension is represented by a risk model, which allows us to model and simulate the system behavior in nominal (failure-free) and degraded (failure) modes [8]. The proposed model is called *Generic Dynamic Agent Based Model* for risk analysis (GDABM).

The remainder of the paper is organized as follows. Section 2 highlights the main dynamic models and Section 3 presents a literature review of the most used methods in risk analysis. Section 4 identifies the selected methods for risk analysis. Section 5 gives a complete description of the classical ABM. Section 6 discusses the proposed model (GDABM). Section 7 has the case study of using the chemical reactor/operator to verify and validate the proposed model. Section 8 has the simulation testbed. Section 9 has the conclusion and the future work of the paper.

## 2. Dynamic Modeling

Dynamic modeling, known as *simulation modeling*, is described mainly using mathematical models. Delany et al. [9] assume that dynamic models (DM) are defined using a set of rules. These rules take the current states as inputs and study how the modeled systems change over time. In this subsection, a taxonomy of the main dynamic approaches applied to complex systems will be shown. Borshchev et al. [10] describe DM as a relationship providing the next state of the studied system based on its present state. Min and Zhou [11] categorized the model variables as follows:

- Non-probabilistic/Deterministic models that use static crisp parameters. They are decomposed into two categories: (1) Single objective models and (2) multi-objective models;
- Probabilistic/Stochastic models that include unknown or random parameters [12,13] where Markov models can be used to model such stochastic events. Those approaches can be further classified as (1) the optimal control theory and (2) the dynamic programming;
- Hybrid models have mixed elements from both the deterministic and the stochastic models. Hybrid models include both the simulation aspects and the inventories theory to cover crisp and uncertain parameters;

Beamon [14] and Labarthe [15] have classified the models based on the used tools (such as economic, analytical, simulation, and organizational approaches). One of the important strengths of dynamical modeling is the ability to illustrate the temporal aspect throughout the simulation. Three modeling approaches of the temporal aspect could be observed in Figure 1: Random Number Models, Continuous Time Models, and Discrete Event Models. Hybrid modeling illustrates the combination of those approaches.

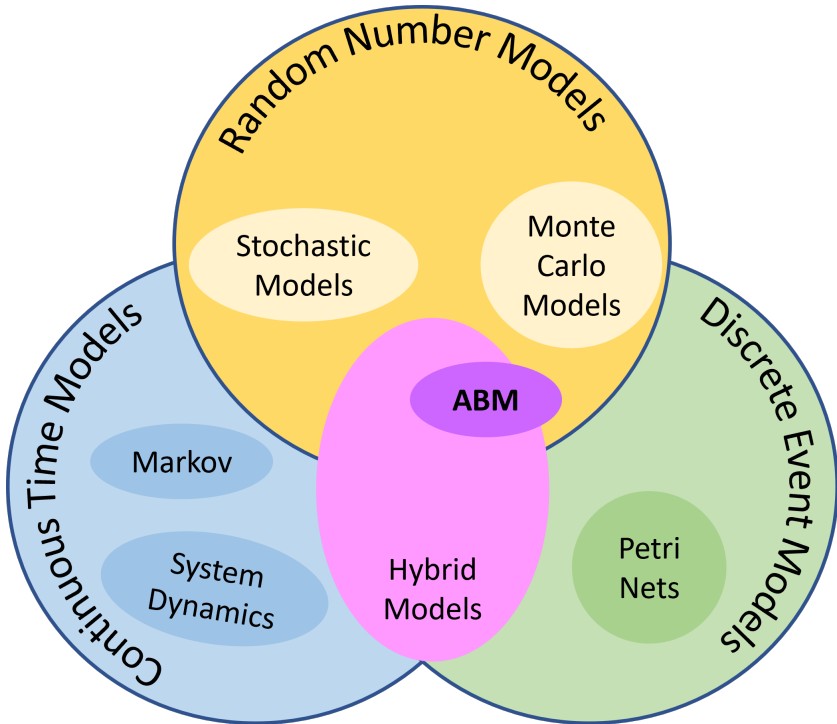

**Figure 1.** Classification of dynamic modeling Approaches.

### 2.1. Random Number Models

In the field of risk analysis, most of the work did not focus on studying the events and hence the probability of detecting events in a simulation is very rare.

A possible solution is to conduct the experiment with a random generation of inputs. A computer is defined as a deterministic machine capable of carrying out instructions fed beforehand, represented as a program. Deterministic algorithms are used for the generation of random numbers (GRN); those numbers should resemble random even on large scales [16]. The best algorithms for GRN have been developed by mathematicians [17]. For reliability analysis, Monte Carlo Models, Markov models, and Agent-based models are on the top of random number models.

### 2.2. Continuous-Time Models

In continuous-time model (CTM) and as the name indicates, a continuous description of the variable's changes is provided using some differential equations. CTM covers:

- System Dynamics (SD): SD is defined as a mathematical model that represents complex systems. The applications of this model are very wide, and it is mainly discrete.
- Markov model: Markov model is a set of consecutive random variables that represent the system evolution dynamically in continuous or discrete-time models. Although Markov chains [18] have been implemented with success in the context of risk analysis [13], they are inadequate for large systems [16], and they are inadequate for short time interval [19].

### 2.3. Discrete Event Models

In contrast to continuous models, for a discrete event system (DES), ref. [20] the state variable changes at discrete/numerous times, with a chronological representation of each operation as a sequence of events. Each event is described by an occurrence time that may change the system's state. For DES, changes may happen only at the moment of event occurrence. DES can be done using activity-based, event-based, three-phase approaches, and process-based [21]. In literature, the most used DES tools are

- Discrete event simulation (DES) describes entity flow and resource sharing using entities, resources, and block charts with the related changes at the prescribed occurrence time [22]. DES is used in different applications and mainly for safety analysis and performance evaluation [23]. Arena, ProModel, Witness, and Anylogic are the most used software for DES.
- Petri Nets (PN) is described as a mathematical modeling language that can represent distributed and discrete systems using some places, arcs, transitions, and tokens. PN was applied successfully in different fields such as reliability analysis, planning of complex production systems, modeling of automated production systems, and management of supply chains [24–29].
- Business Processes Model (BPM): Known as BPMN (Business Process Modeling Notation). It is a standard method representing processes using simple diagrams easily managed by IT and business managers [30,31].

### 2.4. Hybrid Models

- Agent-based modeling (ABM): ABM is a new approach used to model distributed and intelligent systems. It is a decentralized model, highly preferred for complex systems and characterized by the diversity of its abstraction level. ABM was tested and used in different application as supply chain [32], air transport [33,34], health and spread of pandemics [35], and evacuation plan in a fire situation with obstacles [36], but its application for risk engineering science is very limited. ABM is considered a simple modeling tool for complex system representation by modeling only the individual units named agents and simulating their interaction to get the behavior of the whole system [16].
- Logical-combinatorial approach (LCA): It is mainly used for supervised and unsupervised dynamic pattern recognition problems. It aims to classify a set of classes as normal or deviated [37]. The majority of the papers developed using LCA focused on three problems: feature selection, supervised classification, and unsupervised classification [38]. This approach can be used to perform dynamic risk analysis as it can show two different categories of behavioral modes: Normal and Abnormal.

### 2.5. Why Agent-Based Model?

ABM is characterized by the definition of behavior at the individual level. In this work, it is used because of the below:

1. It is made up of several intelligent agents that communicate and cooperate with each other within a distributed and dynamic environment [7];
2. Its intelligence is represented by the ability to make decisions under incomplete/partial perception of its environment [39]
3. Its capability to analyze complex models with a high level of inter-dependencies;
4. Its ability to deal with decentralized/distributed components;
5. Its flexibility: represented by the dynamic number of agents in the simulation;
6. Its ability to detect the unexpected behavior of a complex system;
7. Its very high Computational power allows users to modulate complex systems with micro details.

## 3. Risk Analysis

Risk Analysis (RA) is the first step in the risk management process that aims to identify risk origin, impacted areas, and probable interventions [40,41]. It is defined as a measure of losses on economic, population, and environmental levels. It is characterized by two aspects that provide an evaluation of the related risk level: likelihood or the expected probability $\psi$ of an event occurrence and the severity or the effect $\eta$ of its undesirable consequences. Four risk levels can be distinguished: high (H), significant (S), moderate (M), and low (L) risk regions, as shown in Table 1. Risk assessment process could be used in any system type, such as insurance systems [42], nuclear industry [43], transportation of ice-covered waters [44], building fire evacuation [45], online shopping transaction [46], and food security system [47].

**Table 1.** Risk Level Classification.

| Likelihood of Occurrence ($\psi$) | Severity of Harm ($\eta$) | | | | |
| --- | --- | --- | --- | --- | --- |
| | Catast-Rophic: Death, Injuries | Serious: Extensive Toxic Release | Moderate: Medical Treatment Required | Minor: First Aid Treatment | Negligible: No Injuries or Illness |
| Very Likely $\psi >= 10^{-1}$ | H | H | H | S | S |
| Likely $10^{-3} <= \psi < 10^{-1}$ | H | H | S | S | M |
| Moderate $10^{-6} <= \psi < 10^{-3}$ | H | H | S | M | L |
| Unlikely $10^{-9} <= \psi < 10^{-6}$ | H | S | M | L | L |
| Rare $\psi < 10^{-9}$ | S | S | M | L | L |

### 3.1. Classification of Reliability-Based Methods for Risk Analysis (RMRA)

RMRA are classified into three categories: Qualitative, Semi-Quantitative, and Quantitative, which depend on the type of available data [48,49]. Those categories with the covered methods are visualized in the form of a Venn diagram, presented in Figure 2.

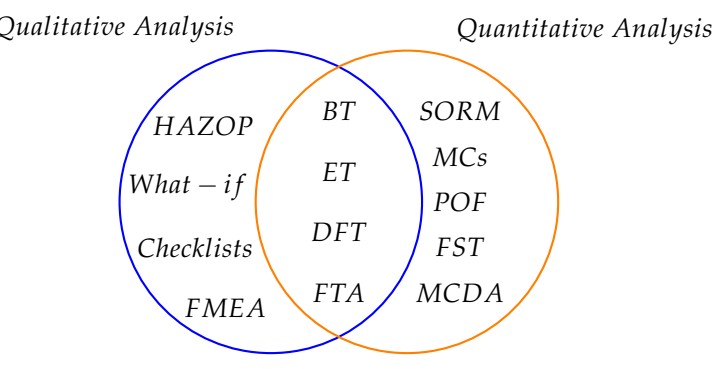

**Figure 2.** A classification of the presented reliability-based methods for risk analysis.

### 3.2. Qualitative Risk Assessment (QRA)

Insufficient data leads to a qualitative risk assessment that uses some information about hazards, causes, and outcomes of failure, in addition to the probability of failure events, to produce reliability. The most used methods for QRA are Hazard and Operability Study (HAZOP) [50], What-if/Checklist, Logic diagrams, and Failure Modes and Effects Analysis (FMEA) [51].

### 3.3. Semi-Quantitative Risk Assessment (SQRA)

SQRA is to be applied whenever the studied system requires further details than the qualitative approach [49]. It covers some quantity of probability, outcomes, and risk value. It may be conducted using fault tree analysis (FTA), Dynamic fault tree (DFT), event tree analysis (ETA), bow-tie analysis (BT), or risk ranking matrix [52].

#### 3.3.1. Dynamic Fault Tree

FTA is used for different applications to study system dependability [53]. It consists of many logic gates that represent the relationship between failures and their origin. In an FTA, basic events are independent, and no consideration of events sequencing/order is possible [54]. This model is called Static Fault Tree (SFT). In literature, many attempts have been reported to overcome these constraints, with consideration of the temporal aspect and statistical dependencies in the FT model. In 1976, Fussell et al. [55] have introduced the concept of Priority-AND (PAND) gate. Later, many other extensions to the SFTs have been proposed (e.g., DFT [56], temporal fault trees [57], and State/event fault trees [58]). The most popular one is the DFT. It retains the PAND gate and adds many others like priority OR (POR), Functional Dependency (FDEP), Warm Spare (WSP), and Sequence enforcing (SEQ). Unlike the static fault tree, DFT uses both Boolean and dynamic gates to specify logical relationships among events.

#### 3.3.2. Event Tree Analysis

An event tree analysis (ETA) is performed following the bottom-up approach. It identifies all potential event sequences which may result from the initial event. Event trees were applied in many cases to analyze risks for chemical processes [59]. This analysis consists of two parts: analyzing the causes of an event (failure mode) using DFTA and identifying the sequence of events using ETA. The combination of DFTA and ETA forms a Bow-Tie Analysis (BTA).

### 3.4. Quantitative Risk Assessment (QRA)

QRA is to be applied whenever the analyzed risks need further detailed analysis. QRA assesses risks to identify and prioritize technology needs and evaluate regulatory alternatives [60]. Quantitative methods used in the literature can be analytical (such as the probability of failure POF, second-order reliability method SORM), probabilistic (Monte-Carlo simulation MCS, stochastic response surface methods SRSM), or sophisticated (fuzzy set theory FST, multi-criteria decision analysis MCDA) [61]. Those reliability-based methods are then categorized into FM analyses (FMEA), tree and diagrammatic analyses [62–65] (FTA, DFT, ETA, and BT), and hazard analyses (HAZOP). Ref. [66] contains further details of risk assessment methods. Complex mathematical and statistical problems can be easily represented and solved using Monte Carlo simulation. It was applied in many fields such as Energy, finance, project management [67], engineering [68], insurance, transportation [69], human health risk assessment [70], and manufacturing. Kolios et al. [71] declare that MCs come with high computational effort, which is considered the main disadvantage.

## 4. Selected Methods for Risk Analysis

After providing an overview of the main methods used for risk assessment, Table 2 highlights some capabilities and limitations of those methods.

**Table 2.** A comparison between the main reliability methods.

| Method | Capabilities | Limitations | Reference |
|---|---|---|---|
| FMEA | Easy implementation | Competent facilitator for reaching consensus in scoring | [51] [72] |
| FTA, ETA | Visual representation of events relations | Cumbersomeness in case of highly granulated analysis | [59] |
| BTA | Efficient link of ETA and FTA | Common cause and dependency failures | [73] [74] |
| Dynamic FTA | Representation of dependent events | Inaccurate results for inappropriate SDE | [75] |
| HAZOP | Structure description of hazard | Extensive documentation | [61] |
| MCS | Direct simulation, easy to implement | Large computational effort | [76] [77] |

As shown in Table 2, Dynamic Fault Tree (DFT) and Monte Carlo simulation (MCs) are the most suitable method for dynamic systems. MCs is considered the ideal solution to model random events with rare probability, which is the case of failure events [78], but this method requires a high computational effort [71]. DFT is the best method to be used for fully dynamic systems with consideration of probability of failure and repair rate [79]. In this work, the authors used DFT to consider the dependencies, sequences, and redundancies of FMs using special dynamic logic gates [73]. Furthermore, it allows the representation of the combination of events and the effects of the order of the failure [79]. In contrast to the static fault tree, DFT covers dynamic and logical gates that represent the relationship between the studied events.

As in this work, the authors aim to represent the dynamic agent's behavioral and failure modes, so DFT was used to represent the failure propagation between the system's components and perform risk analysis.

## 5. Classical Agent-Based Model (ABM)

This section discusses the classical ABM modeling technique [5]. ABM models the agents of a specific system and simulates the interactions of these agents with the environment to get the overall system behavior, as shown in Figure 3. ABM is used to model and simulate systems in different sectors [80] such as traffic [81], Epidemic transmission (COVID-19) [82], and construction [83]. The following subsections discuss the agents and the environment modules in addition to the use of ABM to assess risk analysis.

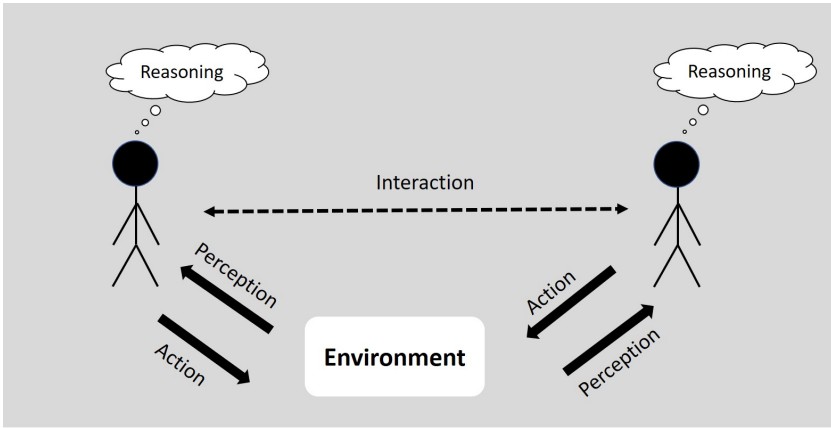

**Figure 3.** Agent interaction with the environment.

*5.1. Agent*

In literature, the term *agent* has many definitions. In this paper, an agent is defined as an autonomous entity with an informative state and could be either software or hardware [84]. The informative state $S$ is defined as in the tuple $S = <X, Y, BMs>$ represented in Figure 3 where:

$X$. It is a finite set $\{x_1, \cdots, x_\theta\}$ of variables that define the dynamic characteristics of an agent, where $\theta$ is the total number of variables for an agent.

$Y$. It is a finite set $\{y_1, \cdots, y_\rho\}$ of attributes that define the static characteristics of the agent, where $\rho$ is the total number of attributes for an agent.

*BMs*. It is a finite set of behavioral modes (BM)=$\{BM_1, \cdots, BM_\rho\}$ that specifies the rules under which the agent acts, where $\varrho$ is the total number of behavioral mode for an agent covering the nominal and failure modes.

*5.2. Environment*

An environment is the place where an agent is located [84]. For each agent $A_i$, there is an environment $\Omega_i$ defined as the set of all objects/agents outside $A_i$. A mutual relationship exists between agent-environment: agents use any information sensed from the environment to make possible decisions whenever needed and they are capable of producing output actions that affect the environment, as shown in Figure 3. Sometimes, the collected information is incomplete. Due to their intelligence, agents will make their decisions in such conditions of uncertainty [39].

*5.3. Risk Analysis Using Classical ABM*

ABM was used for risk analysis in various fields such as reinforcement learning [85], financial risk [86], social risk [87], oil sector [7], gas sector [88], natural disaster and emergency systems [89–92], disease propagation stochastic modeling systems [93], supply chains [94], intrusion detection and prevention systems for Android mobile devices [95], green edge computing systems [96], and cloud computing [97].

Meanwhile, we have investigated state of the art in multi-agent work that took stochastic systems into consideration and we have listed the following references: smart electricity grids and markets, biology epidemics distribution systems, and ecological systems [98–103]. However, they didn't show the details of the methodology used in the problem-solving process. A limited number of authors are explicitly using ABM as a novel modeling approach [104–106] and their proposed approach does not represent the failure propagation between system agents. To do so, a risk model should be considered for ABM. This model is presented in Section 5.

## 6. Generic Dynamic ABMS (GDABM) for Risk Analysis

This section presents a proposed extension of ABM allowing the representation of the overall system behavior in normal and degraded modes in addition to the analysis of existing risks. This extension forms the new risk model called *Generic Dynamic Agent Based Model* (GDABM) for risk analysis. Figure 4 shows the 4 components of GDABM:

1.   Behavioral Modes (BMs);
2.   Failures Modes (FMs);
3.   External Failure Agent Communication (EFAC);
4.   Internal Failure Agent Communication (IFAC).

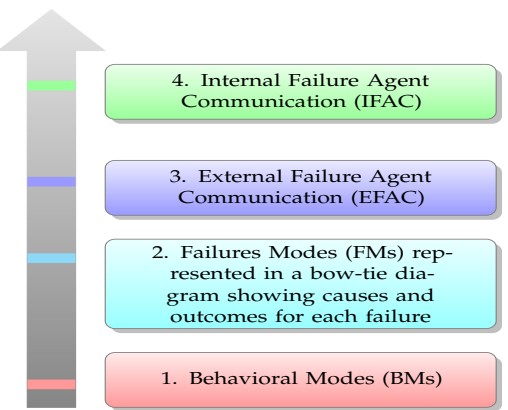

**Figure 4.** Proposed features to be added for the classical agent model.

Those components provide a standard pattern (or metamodel) for the system' agents. The first component is used to illustrate the system in its different operating modes. The second component allows risk situations to be identified with full details and components 3 and 4 are used to assess and represent the spread of risk from one element to another. GDABM represents the contribution of this work because these four components are the elements necessary and sufficient to model any agent behavior and also analyze and assess the risk according to the evolution of the system. The following sections discuss the components.

### 6.1. Behavioral Modes (BMs)

In the field of risk analysis, the concept of agent mode defines the agent's operational behavior in the presence of failure conditions. In the same way, the *nominal agent mode* defines the agent's operational behavior without the presence of any failure. *Behavioral modes* (BM) define the agent's behavior in both its nominal and degraded modes. BM describes the dynamic behavior of a multi-agent system by continuously measuring the behavior of each agent in that system. As cited in Section 4, an agent is defined by a set of variables, attributes, and behavioral modes. Its dynamic movement in a behavioral mode $M_i$ is defined using a set of sequential modules/blocks represented as activity blocks. Those modules can be of 4 types, as shown in Figure 5.

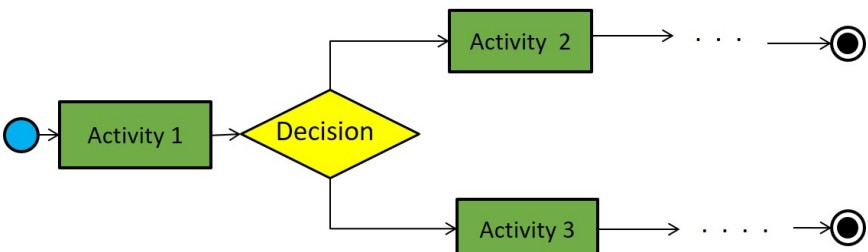

**Figure 5.** Behavioral mode of an agent.

1. Start Module (SM) : It is the starting point at which the agent is created and ready to process.
2. Activity Process Module (APM) : It represents the various activities to be processed by an agent **a**. It describes the interaction between a and other system agents. Such activities may include creating new agents or deleting existing ones. APM has the following characteristics, as shown in Figure 6.
   (a) A mathematical relation : It can be of two types:
      - Discrete relations $f_i$:

$$x(k+1) = f_i(x(k), y(k), u(k), v(k)), M(f_i); \qquad (1)$$

- Continuous relations $g_i$:

$$x^*(k) = g_i(x^*(k), y(k), u(k), v(k)), M(g_i);\qquad(2)$$

where: $x(k)$: finite set of agent **a** variables;
$y(k)$: finite set of agent **a** attributes;
$u(k)$: variables of agents in relations with **a**;
$v(k)$: attributes of agents in relations with **a**;
$M(f_i)$: set of behavioral modes $M(f_i) \subseteq BM$, when $f_i$ is valid;
$M(g_i)$: set of behavioral modes $M(g_i) \subseteq BM$, when $g_i$ is valid;
$x^*$: subset $x^* \subseteq x$ of the agents variables;

(b) A duration : It is the time required to execute the activity process;
(c) Consumable Input Agents: They are consumable agents that help to generate the output agents.
(d) Non-consumable Input Agents : They are non-consumable agents that should be allocated to the agent activity to perform a certain task then get released on task completion.
(e) Output Agents: They are the agents produced at the end of the activity.
(f) Activity Agent: It is the agent executing the activity.
(g) Activity engine: It is the core of the activity process that identifies the inputs/outputs agents and controls the actions among different activity components. It describes how to generate output agents using input agents.
(h) Inputs Actions: They are pre-actions that should be performed just before the execution of the APM (e.g., allocating non-consumable agents for a certain amount of time)
(i) Outputs Actions: They are post-actions that should be performed once the APM is performed (e.g., deallocating non-consumable agents after the task is completed).
(j) Filtering Conditions: Which precise the criteria required for consumable/non-consumable agents of the activity.

3. Decision Making Module (DMM): It is the module responsible for checking some conditions on the agent's variables. The result decides how the agent proceeds.
4. End Module (EM) : It is the point where the agent is terminated and deleted from the system.

Making a coffee represents an example of the Activity Process Module. In the coffee preparation process, the following assumptions are used:

- Duration is the amount of time to make a cup of coffee which is assumed to be 45 s.
- Consumable input agents are coffee powder, water, electricity, and an empty cup.
- Non-consumable input agents are coffee room and the coffee table.
- Output agent is the prepared cup of coffee.
- Input action is the process of reserving the coffee machine/making the water temperature 65.
- Output action is the process of releasing the coffee machine.
- Filtering Conditions is the process of selecting one coffee powder brand among a set of alternatives in the kitchen.

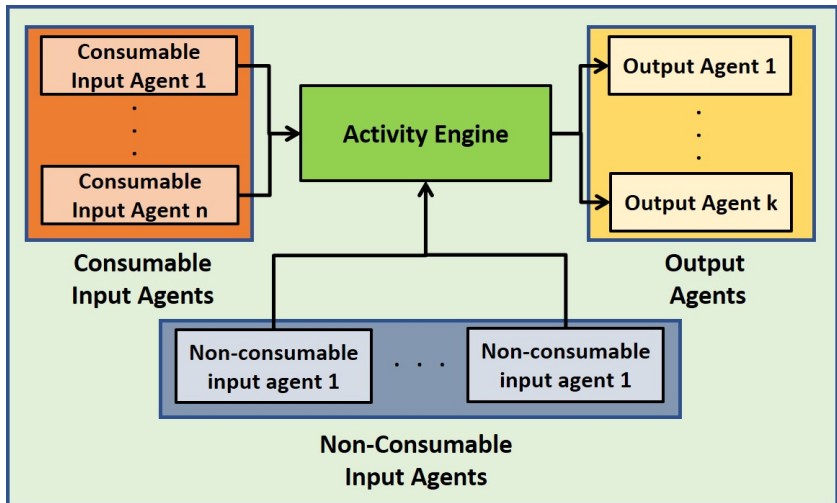

**Figure 6.** An Activity Process Module.

*6.2. Failure Modes (FMs)*

Failure modes (FMs) are events that describe the agents' failures in detail. For each agent in the system, FMs identify (1) what triggers the agents to fail and (2) what caused the agent's inability to comply with the expected level of performance. FMs assume the following:

- Facts are represented by events;
- Agents can have one or more events.
- An event can be either active or inactive;

Failure modes have different attributes as described in Equation (3). They are classified into three categories, Boolean Failure Modes *BFMs*, Stochastic Failure Modes *SFMs*, and Complex Failure Modes *CFMs*.

$$FM =< N, A, F, S > \tag{3}$$

where $N$, $A$, $F$, and $S$ are:

- $N$ is the failure mode's name;
- $A$ is the agent that experiences the failure;
- $F$ is the current value of the failure whether it is active or inactive failure.
- $S$ is the set of successor events in case of active failure, represented in an event tree.

1. Boolean Failure Modes (BFMs): A Boolean Failure Mode is an event representing a certain condition/expression (e.g., a > b, a + b < c, $\cdots$) and has the value of that expression. Once this expression is true or valid, the failure mode is said to be active. In general, the expression is directly related to the agent's variables. *BFM* is expressed in terms of the Boolean expression $B$ as in Equation (4):

$$BFM =< N, A, F, S, V > \tag{4}$$

where $V$ is the Boolean expression associated with the agent variable(s).

2. Stochastic Failure Modes (SFMs): *SFM* is a failure mode defined as a probability of failure. *SFM* is represented in Equation (5)

$$SFM =< N, A, F, S, P > \tag{5}$$

where $P$ is the probability that represents the likelihood of the system's failure;

3. Complex Failure Modes (CFMs) A *CFM* is defined as in Equation (6):

$$CFM =< N, A, F, S, D > \tag{6}$$

where $D$ represents the set of predecessor events of the *CFM* represented in a dynamic fault tree. Predecessor events could be either of type *BFM* or *CFM*. A *CFM* is enabled when the result of the output of the combinational circuit is enabled.

Figure 7 has the bow-tie diagram. It consists of the combinational circuit, PAND, POR, SEQ, SPARE and FDEP gates, representing the dynamic fault tree. The diagram also has the event tree representing the set of consecutive events that occur on failure. The bow-tie diagram is a typical example of the CFM.

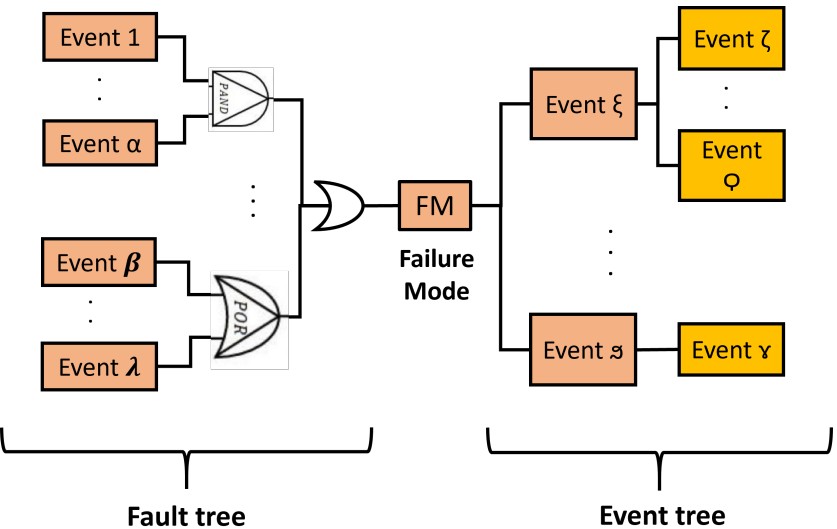

**Figure 7.** A Bow-Tie Diagram Activity.

### 6.3. External Failure Agent Communication (EFAC)

An External Failure Agent Communication (EFAC) governs the communication among different agents (connection **d**). A failure of an agent $i$ could be propagated to other surrounding agents. When a failure mode $i_{FM}$ of an agent $i$ becomes active, this agent broadcasts a message to all surrounding agents. The propagated message contains complete information about the failure. This failure will be added to the set of external failure elements of the surrounding agent's failure modes.

For example, if we consider a multi-agent environment where the agents are trucks moving in a highway. If a truck $t1$ travels from a point A to a point B, a collision between two other trucks $t2$, and $t3$ in the same path of the truck $t1$, might cause a significant delay to truck $t1$. This collision information will be shared with $t1$ and it is considered as an external agent failure for $t1$.

### 6.4. Internal Failure Agent Communication (IFAC)

In the proposed GDABM model, there is a bidirectional influence between the FMs and the BMs for any agent. This influence describes how the change in the value of the agent's variables in a behavioral mode might trigger an agent's failure mode.

For example, in a car, many failure modes could occur. Failure modes could be mechanical, electrical, fuel-based, car body, etc. Initially, all of these failures are assumed to be inactive. The car, in this case, is assumed to be functioning properly (in its nominal mode). In case of any failure activation to any of the aforementioned components, that would lead to a degraded functionality of the car (degraded mode) and might lead to a more severe total dysfunctional of the car.

There are two sets associated with any agent $i$, BMs set ($i_{BM}$) and FMs set ($i_{FM}$). If the number of elements in the FM set is $\mu$, then the number of elements in the BM set can take up to $2^\mu$ values, one of which is considered nominal .

The following subsections have the influence of the BMs on the FMs and vice versa.

### 6.4.1. BMs → FMs

This section has the influence of the BMs on the FMs (connection **a**). The BMs are assumed to have a set of variables denoted by $X$. These variables have constraints. The constraints on the agent's variables define a set of CBFM for that agent.

Equation (7) computes the FM as a function of the BM's variables.

$$i_{FM} = \varphi_i(X) \tag{7}$$

where: $i_{FM}$: is a CBFM of the agent i, $\varphi_i(X)$: is a boolean expression of $X$ for an agent $i$. If $\varphi_i(X)$ is true, $i_{FM}$ becomes active and will be added to the set of active failure modes of the agent $i$.

### 6.4.2. FMs → BMs

The behavior of an agent $i$ in a multi-agent environment is assumed to be initially in its nominal mode $i_{Nom}$.

Nominal mode $i_{Nom}$ contains a set of activities $v$. Each activity $v$ has a set of FMs. The set $i_{IFAC}$ covers all possible FMs that are generated within the agent during the execution of any activity $v$. Moreover, $i_{EFAC}$ covers the set of all possible failure modes that occur by other external agents.

For each agent $i$, a set of failure modes $i_{FM}$ is defined as an in Equation (8):

$$i_{FM} = i_{IFAC} \cup i_{EFAC} \tag{8}$$

Equation (9) computes the behavioral mode $i_{BM}$ of an agent $i$ as a function $\vartheta$ of the set of the active failure modes $i_{FM}$ of that agent.

$$i_{BM} = \vartheta(i_{FM}) \tag{9}$$

If $i_{FM}$ does not contain any FM elements, $i_{FM} = \phi$, then the $i_{BM}$ of the agent $i$ is nominal $i_{Nom}$. On the other hand, any addition of a failure mode element to the set of active failure modes leads to a disruption of the behavioral mode (connection **b**).

Figure 8 represents the proposed model (GDABM). The figure shows the interaction between an agent i and its environment. The GDABM is composed of: (1) Risk Model Block (FM): It has the set of failure modes available in the GDABM in addition to their causes and consequences. Sources/causes of failure modes are described in a dynamic fault tree and their consequences are illustrated in an event tree. For each failure mode, there is an associated behavioral mode to be triggered (2) Behavioural Mode (BM) module contains a set of degraded modes that are possible to occur in addition to the nominal mode. (3) Set of Variables: It holds the static characteristics of the agent. (4) Set of Attributes: It holds the dynamic characteristics of the agent. (5) Agent's environment: It holds the external agents.

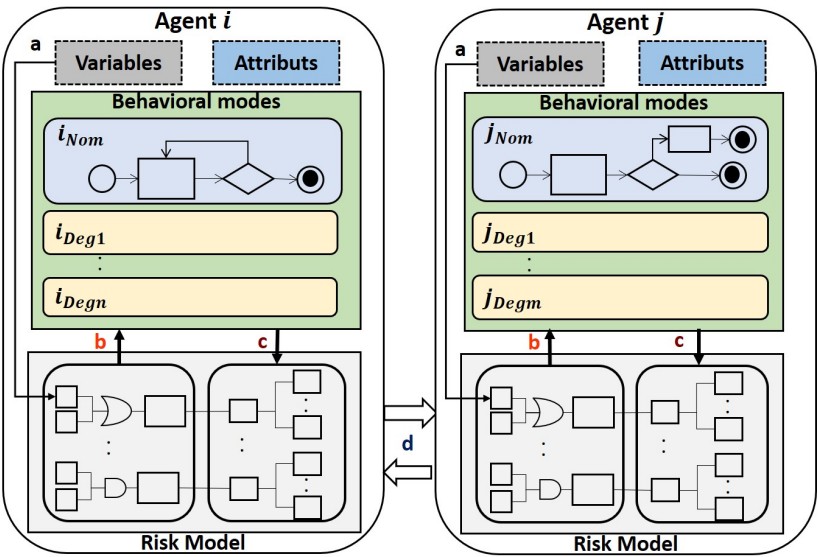

**Figure 8.** Generic Dynamic Agent-Based Risk Model (GDABM).

## 7. Case Study: Modelling Chemical Reactor/Operator Using GDABM

This section has a detailed case study of a multi-agent system that uses the proposed *Generic Dynamic Agent-Based Model* (GDABM) for risk analysis to models and simulates both nominal and degraded conditions of a chemical reactor system that is widely used in the industry [107].

The chemical reactor takes input products from two different production lines, $ProductionLine_1$ and $ProductionLine_2$. The chemical reactor mixes the two products together in a chemical reaction resulting in an output product. The output product is placed in a third production line $ProductionLine_3$. The chemical reactor system consists of two main agents *agent reactor* and *agent operator* as shown in Figure 9.

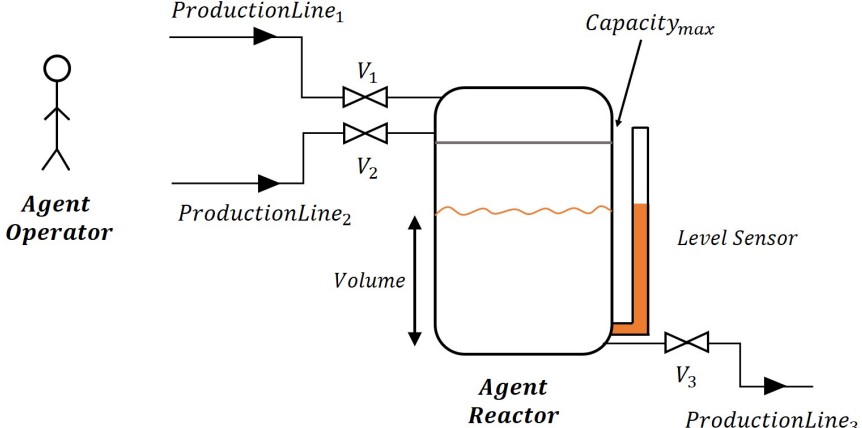

**Figure 9.** Chemical Operator/reactor.

### 7.1. Agent Reactor

The *agent reactor* has production lines $ProductionLine_1$ and $ProductionLine_2$.

The agent reactor is connected to the three valves ($v_1$, $v_2$, and $v_3$). Valves $v_1$ and $v_2$ are the input valves used to load products to the reactor. Valve $v_3$ is used to unload the products. The reactor is equipped with a level sensor that reads the current *volume* of the product inside the reactor in real-time. During a chemical reaction, the reactor enters in a state lock then it will remain unlock once the reaction is done.

The agent reactor has one attribute ($V_{max}$) that represents the maximum capacity of the reactor in addition to seven variables that are described as follows:

1.  *Volume V*: It has the current volume of the product in the reactor,
2.  *Gas Concentration* (*GC*): It has the concentration of the gas in the reactor's environment,
3.  *Release Rate RR*: It is the rate in which the gas is released from the reactor.
4.  Input $Iv_1$: It is the valve used to load the products from *ProductionLine*$_1$ when $v_1$ is open.
5.  Input $Iv_2$: It is the valve used to load the products from *ProductionLine*$_2$ when $v_2$ is open.
6.  Output $Ov_3$: It is the valve used to unload products to *ProductionLine*$_3$ when $v_3$ is open.
7.  State *S*: It describe the state of the reactor that can be **Locked L** or **Unlocked U**.

The reactor's nominal mode has two activities *transform products*, and *wait*. Transform products is the activity that transforms two quantities of consumed elements, $P_1$ and $P_2$, to produce $P_3$ (produced element), as shown in Figure 10. The $transform - product$ activity is only enabled when the reactor has products ready and its state is Locked. In the nominal mode, whenever the state of the reactor is Unlocked, Wait activity is triggered.

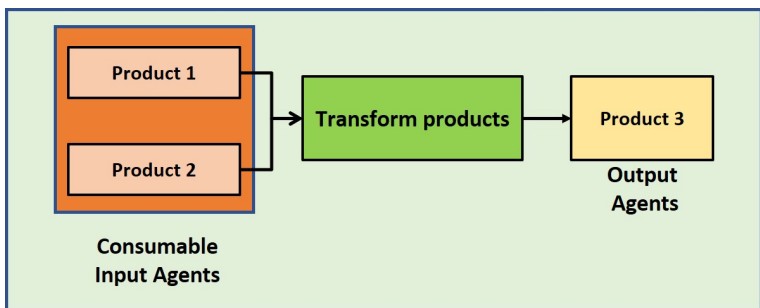

**Figure 10.** Transform-Product Activity.

*7.2. Agent Operator*

The *agent operator* has two attributes. The first one is the *Gas Concentration Threshold* ($\gamma$) which is the maximum value of the gas concentration above which it is considered to be toxic and needs immediate attention. The second one is the *exposureTime* ($\tau$) that represents the maximum exposure time of an operator to a toxic Gas release before being out of order (irreversible state). It has also four variables:

1.  Input $P_1$: It is the maximum quantity of products to be loaded from *ProductionLine*$_1$.
2.  Input $P_2$: It is the maximum quantity of products to be loaded from *ProductionLine*$_2$.
3.  Output $P_3$: It is the maximum quantity of products to be unloaded from the reactor.
4.  State *S*2: It describe the state of the reactor that can be Idle , Inactive , or Out of order.

Initially, the operator and the reactor agents are assumed to be functioning properly in their associated nominal modes.

The operator's nominal mode has four activities:

1.  Load: The load activity is the process of filling the reactor's production lines *ProductionLine*$_1$ and *ProductionLine*$_2$ with quantities $P_1$ and $P_2$ respectively.
    The products' incoming rates to the production lines are assumed to be $d_{v_1}$ and $d_{v_2}$ respectively.
    The load activity is executed with consideration of the following: the total quantity of the products to be added to the reactor ($P_1+P_2$) in addition to the quantity of products inside the reactor ($V$) is less than or equal to $V_{max}$ as shown in Equation (10).

$$P_1 + P_2 + V <= V_{max}. \tag{10}$$

2.  Unload: The unload activity is the process of pumping out an amount $P_3$ through *ProductionLine*$_3$ with outgoing rate $d_{v_3}$.
3.  Wait1: This activity represents the process of waiting for the reactor to be Unlocked. It is a pre-process of the load activity in a Locked reactor.

4.  Wait2: This activity represents the process of waiting for the chemical reaction to be performed in time $\tau$. It is an intermediate process between the load and the unload activities.

The Load/Unload activities are only enabled when the state of the reactor is Unlocked. In the nominal mode, whenever the state of the reactor is locked, a Wait activity is triggered.

### 7.3. Failure Analysis of the Chemical Reactor/Operator Using GDABM

Figure 11 illustrates the Reactor/operator system with the proposed risk model that shows for every agent the set of all possible failure (including fault and event trees), and behavioral modes plus the mutual relation between them.

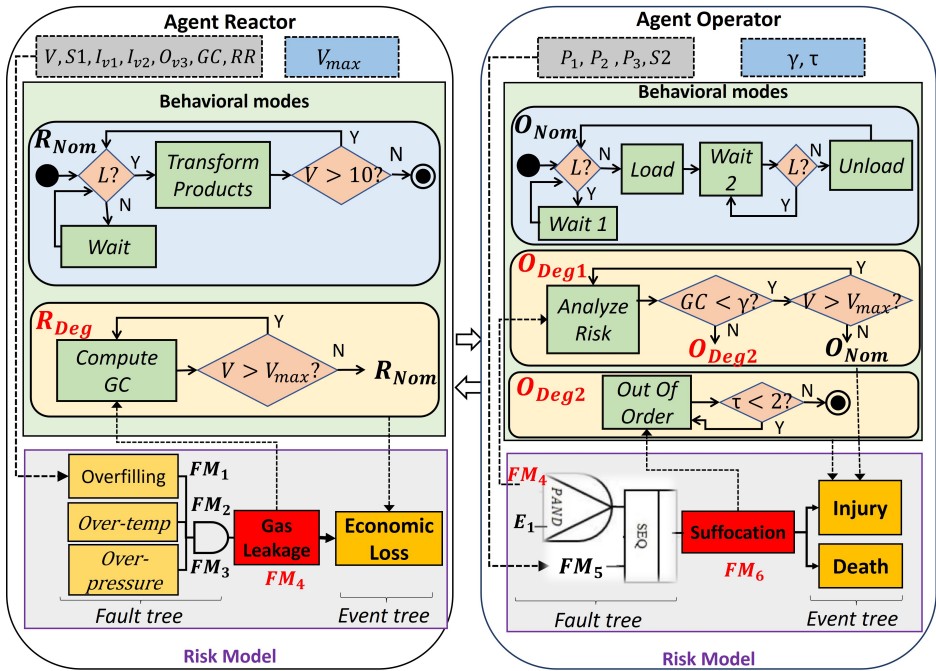

**Figure 11.** Reactor/Operator with the risk model.

The Agents reactor/operator experience different failure events and failure modes. Table 3 contains seven different failure modes that are used as examples in this paper. The first failure mode $FM_0$ illustrates quantity above threshold in the reactor caused by the *Transform products* activity or a misread of the level sensor. The second failure $FM_1$ is overfilling that take place with the existence of a malfunctioned operator and level sensor failure followed by a quantity above threshold Figure 12.

$FM_2$ represents Over-temperature and $FM_3$ Over-pressure. The top event for the agent reactor is $FM_4$ that represents gas leakage from the reactor. $FM_4$ occurs when at least one of the failure modes $FM_1$, $FM_2$ and $FM_3$ occur.

**Table 3.** Agents failure modes.

| Failure Mode | Type | Agent | Description |
| --- | --- | --- | --- |
| $FM_0$ | Boolean | Reactor | Quantity above threshold ($V > V_{max}$) |
| $FM_1$ | Complex | Reactor | Overfilling |
| $FM_2$ | Complex | Reactor | Overtemperature |
| $FM_3$ | Complex | Reactor | Overpressure |
| $FM_4$ | Complex | Reactor | Leakage |
| $FM_5$ | Boolean | Operator | Toxic inhalation ($GC > \gamma$) |
| $FM_6$ | Complex | Operator | Suffocation |

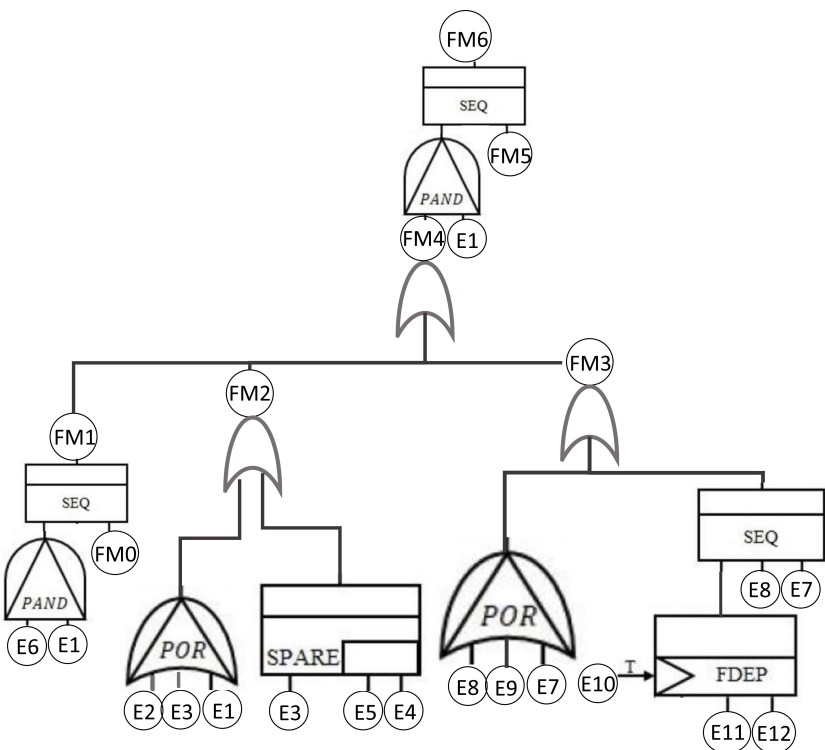

**Figure 12.** Dynamic Fault Tree.

By consequence, reactor' behavior mode changes from $R_{Nom}$ (*nominal mode*) to $R_{Deg}$ (*degraded gas leakage mode*) where the gas concentration is measured continuously through an activity called *compute GC (gas concentration)*, and the agent reactor sends a *gasleakage* message to the agent operator. $FM_4$ is considered as an external failure mode to the operator *operator*$_{EFAC}$, as the failure occurs at the reactor rather than at the operator. Once $FM_4$ occurs, the behavioral mode of the agent Operator change form $O_{Nom}$ (*nominal mode*) to $O_{Deg1}$ (*degraded mode*), where the operator evaluates the risk level related to the Gas Leakage.

$FM_5$ is the failure mode of the operator representing a gas concentration $GC$ exceeding the *Gas Concentration Threshold $\gamma$*. $FM_5$ is an internal failure mode *reactor*$_{IFAC}$. $FM_5$ is BFM; since it depends on the condition whether the gas concentration exceeds the threshold value or not. $FM_6$ is a suffocation failure mode that occurs if the sequence of $FM_4$, $E_1$ and $FM_5$ is valid. Once $FM_6$ is enabled, operator' behavior mode changes from $O_{Deg1}$ (*operator's degraded mode 1*) to $O_{Deg2}$ (operator's degraded mode 2), where the state of the operator change from inactive to out of order due to the toxic inhalation.

Table 4 summarize the main events that might occur during the chemical reaction including their descriptions and probability of failure [108].

Those events and the related failure modes are then represented in a dynamic fault tree as shown in Figure 12.

A gas leakage eventually causes evaporation of Hydrogen sulfide H2S that reduces the volume level inside the reactor. The atmospheric dispersion of gases continues until the failure is fixed or the volume of the product becomes less than the *capacity*$_{max}$ and hence it might eventually restore its nominal mode.

Table 5 represents the various activities and their associated equations where $V^+$, $GC^+$ represent the products' volume and the gas concentration in the environment of the reactor at the next time step (t+1), respectively. The activities *ComputeGC*, *AnalyzeRisks*, and *OutOfOrder* are to be executed with failure presence by the agents Reactor/Operator as shown in Figure 11.

**Table 4.** Description of the main events and their corresponding failure probabilities.

| Event | Description | Probability | Source |
|---|---|---|---|
| E1 | Operator failure of abnormal situations cognition | $2.11 \times 10^{-3}$ | Expert |
| E2 | Failure of the temperature controller | $3.52 \times 10^{-4}$ | Historical data |
| E3 | Over temperature in work environment | $1.38 \times 10^{-2}$ | Expert |
| E4 | Operator fails to shut down the reactor due to over temperature | $4.52 \times 10^{-2}$ | Expert |
| E5 | Air cooling system failure | $8.94 \times 10^{-2}$ | Expert |
| E6 | Level sensor failure | $3.54 \times 10^{-2}$ | Expert |
| E7 | Operator fails to shut down the reactor due to over-pressure | $2.67 \times 10^{-2}$ | Expert |
| E8 | Over pressure in the reactor due to blockage | $1.45 \times 10^{-2}$ | Expert |
| E9 | Pressure controller failure | $3.52 \times 10^{-4}$ | Historical data |
| E10 | Power supply failure | $8.36 \times 10^{-2}$ | Expert |
| E11 | Failure of the steam supply | $1.43 \times 10^{-2}$ | Expert |
| E12 | Valve failure | $6.80 \times 10^{-6}$ | Historical data |

**Table 5.** Activities equations.

| Activity | Equation | Input | Output | Duration |
|---|---|---|---|---|
| Load | $V^+ = V + P_1 + P_2$ | $I_{v1} = I_{v2} = 1$ | $I_{v1} = I_{v2} = 0$ | 1 |
| Transform-product | $V^+ = V * 1.1$ | S1 = lock | S1 = unlock | 5 |
| Unload | $V^+ = V - P_3$ | $O_{v3} = 1$ | $O_{v3} = 0$ | 1 |
| Compute Gas Concentration | $GC^+ = GC + RR$ $V^+ = V - 2$ | S1 = lock | S1 = unlock | 1 |
| Analyse Risks | $RL = f(\psi, \eta)$ | S2 = Idle | S2 = Inactive | 1 |
| Out Of Order | | S2 = Inactive | S2 = OutOfOrder | 2 |

Figures 13 and 14 represent the behavioral modes transitions for the agents Reactor and Operator.

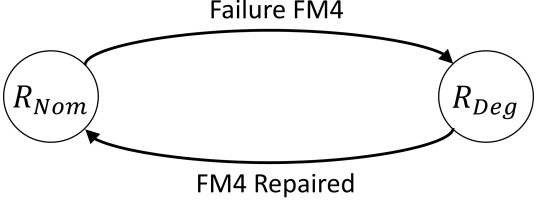

**Figure 13.** Behavioral modes of the Agent Reactor.

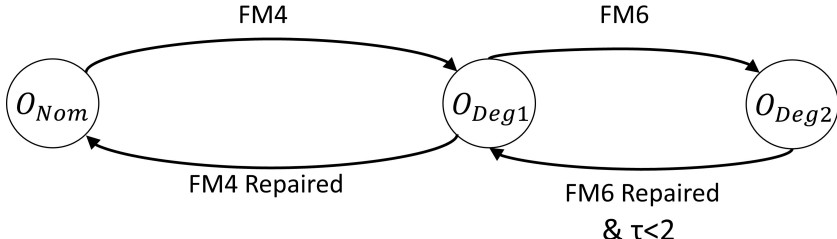

**Figure 14.** Behavioral modes of the Agent Operator.

## 8. Simulation Testbed

In literature, many software languages and tools specifically focused on ABMS development have become established as open-source ABMS platforms, such as NetLogo, Swarm [109], AnyLogic, Repast [110], JADE [111], and MASON [112]. A comparison between those simulators is presented in Table 6 according to many criteria.

**Table 6.** Comparison between GDABM and reference simulators.

| Parameter | Swarm | Repast | Mason | GDABM |
|---|---|---|---|---|
| License | General Public Licence (GPL) | GPL | GPL | GPL |
| User Base | Diminishing | Large | Increasing | Large |
| Execution' Speed | Fast | Moderate | Fastest | Moderate |
| Graphical user interface (GUI) | Limited | Good | Good | Good |
| Built-in ability to create movies and animations | No | Yes | Yes | Yes |
| Easy of learning, programming | Poor | Moderate | Moderate | Moderate |
| Geographical information system (GIS) | Yes | Yes | Yes | Yes |
| Full detailed Risk analysis | No | No | No | Yes |
| Failure analysis | No | No | No | Yes |
| Behavioral modes Identification | Yes | Yes | Yes | Yes |

GDABM was simulated using the Repast Simphony Simulator tool.

The simulation of this model provides the following additional features: (1) identifying and analyzing the risk among system components, (2) studying the risk propagation among these components, and (3) performing the risk evaluation process.

### 8.1. Simulation Results

The simulation results of the proposed model, when tested on the chemical reactor/operator case study discussed in Section 6 are presented in this section. Those results include a representation of the dynamic behavior of each agent in the studied system in addition to the resulting risk level.

Two agents were defined in the above case study: operator and reactor, with their full characteristics including (attributes, failure, and behavioral modes). A simulation of the

chemical reactor/operator system was carried out for a duration of 45 simulation steps to show the dynamic behavior of the various agents in the system.

To test the functionality of the GDABM, experiments were conducted using four different values of the set ($V_{max}$, *RR*) as follows: configuration 1 (30, 10,000), configuration 2 (30, 15,000), configuration 3 (20, 10,000), and configuration 4 (20, 15,000) , and considering the following assumptions:

1.  when a gas leakage occurs, the *volume* is to decrease by 2 L and the gas concentration *GC* is to increase by 10,000 part per million (ppm) at each step of the simulation.
2.  the initial values of the variables are as follow: *Volume* = 10, *GC* = 0, $\tau = 2$, $P_1 = 7$ L, $P_2 = 5$ L and $P_3 = 4$ L.
3.  once the leakage is repaired, the gas concentration is to decrease by 5000 ppm at each step of the simulation.

The ranking scale of the severity of harm $\eta$ is represented in Table 7.

**Table 7.** Threshold limit values for GC.

| GC ($\times 10^3$ ppm) | $\leq 10$ | [10:20] | [20:30] | [30:40] | $\geq 40$ |
|---|---|---|---|---|---|
| Severity | Negligible | Minor | Moderate | Serious | High |

As the likelihood of the Gas Leakage failure $\psi$ is 0.02 (using failure data cited in Table 4), which is between $10^{-3} \leq P < 10^{-1}$, it is considered as likely and the Risk level is evaluated in a dynamic way using the likelihood and the severity values as mentioned in Table 1.

Agent's behavioral modes with the related risk level are represented graphically in Figures 15–18 for the configuration C1, C2, C3, and C4, respectively. Tables A1 and A2 in Appendix A show agents' behavioral modes and the risk level values during the simulation of C1/C2 and C3/C4, respectively. For configurations C1/C3, the release rate is assumed to be 10,000 and $V_{max}$ is 30 for C1 and 20 for C3. The overall risk level is Moderate in C1 except for the time interval [17, 18]; it increases to be significant and the behavioral mode of the Operator agent is $O_{Deg2}$. On the other hand, for C3, as we decreased $V_{max}$ to be 20, which reduces the amount of released materials, the risk level remains Moderate even with the existence of failure events. For C2/C4, the release rate is assumed to be 15,000 and $V_{max}$ is 30 for C1 and 20 for C4. Risk level reached High in C2 for the time intervals [16, 17] and [33, 34]. Those tables represent the failure propagation between agents and the dynamic agents' behavior throughout the simulation.

The chemical reactor/operator multi-agent system was successfully modeled and simulated under the seven failure modes. The GDABM was able to study the dynamics of the various failure mode through risk analysis and risk assessment. GDABM also studies the correlation between agent failure and behavioral modes. GDABM has successfully shown how a change in the agent failure mode affects its behavioral mode and vice versa.

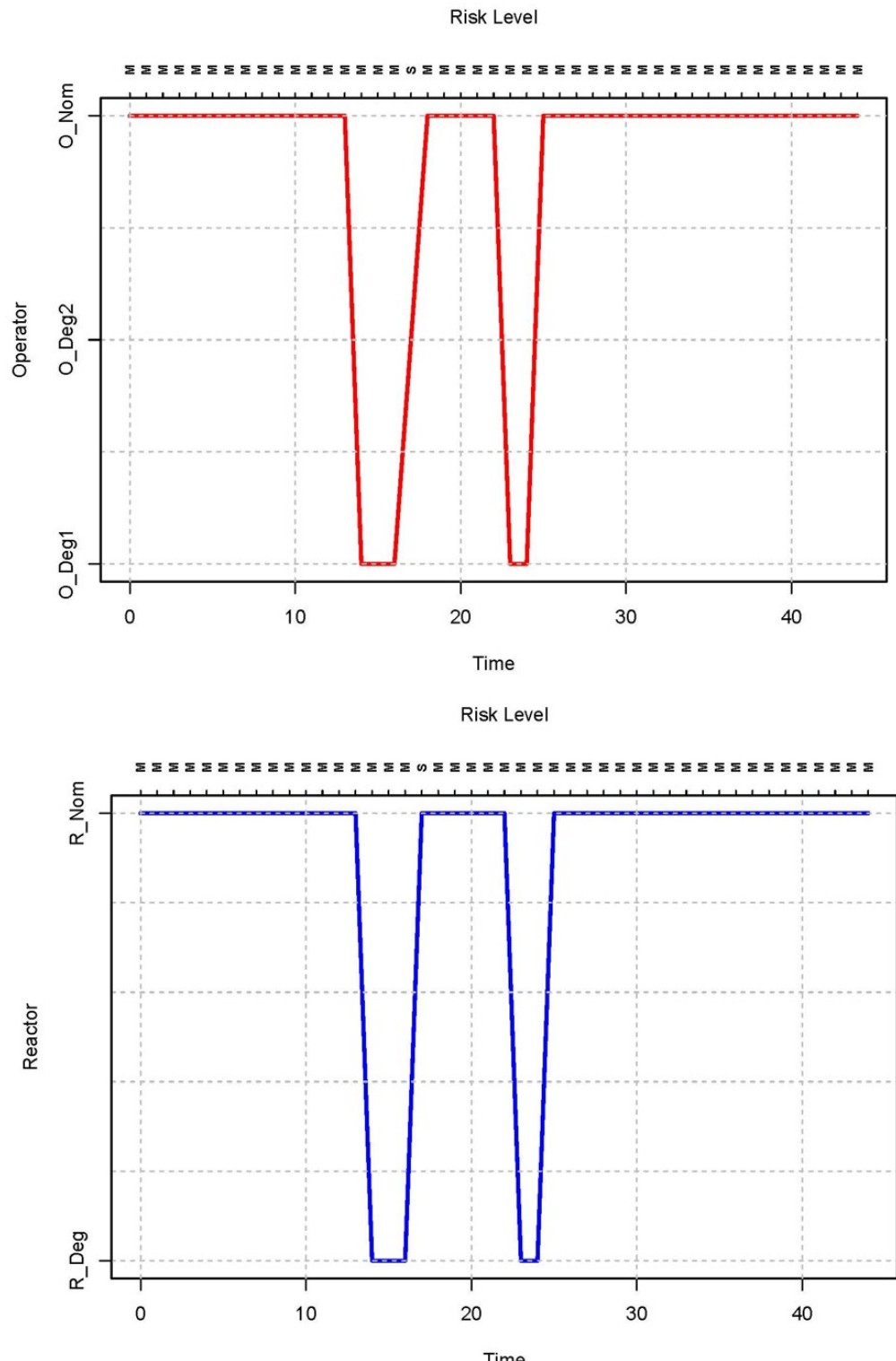

**Figure 15.** Agent's behavioral modes with the related Risk Level during simulation C1.

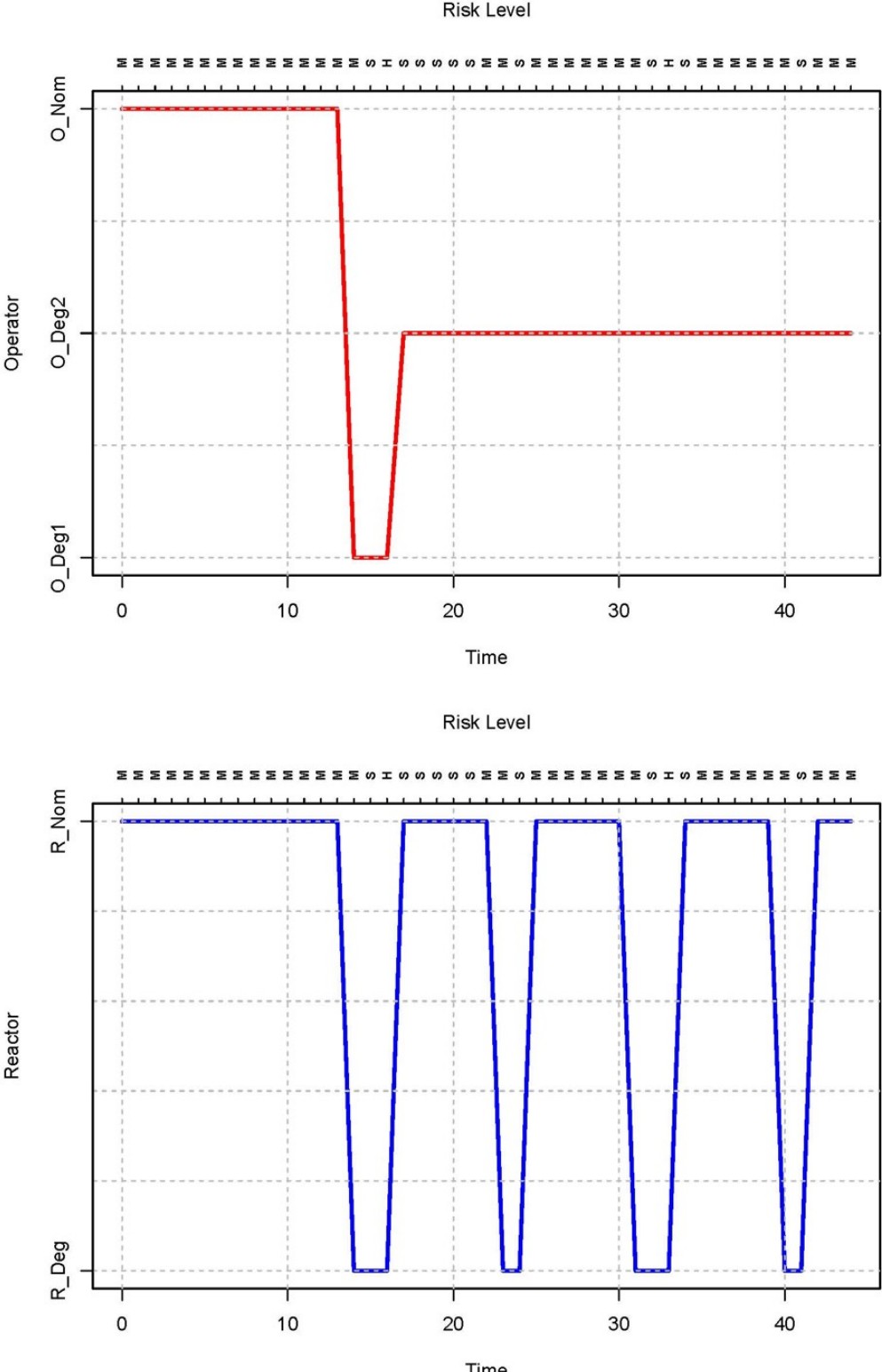

**Figure 16.** Agent's behavioral modes with the related Risk Level during simulation C2.

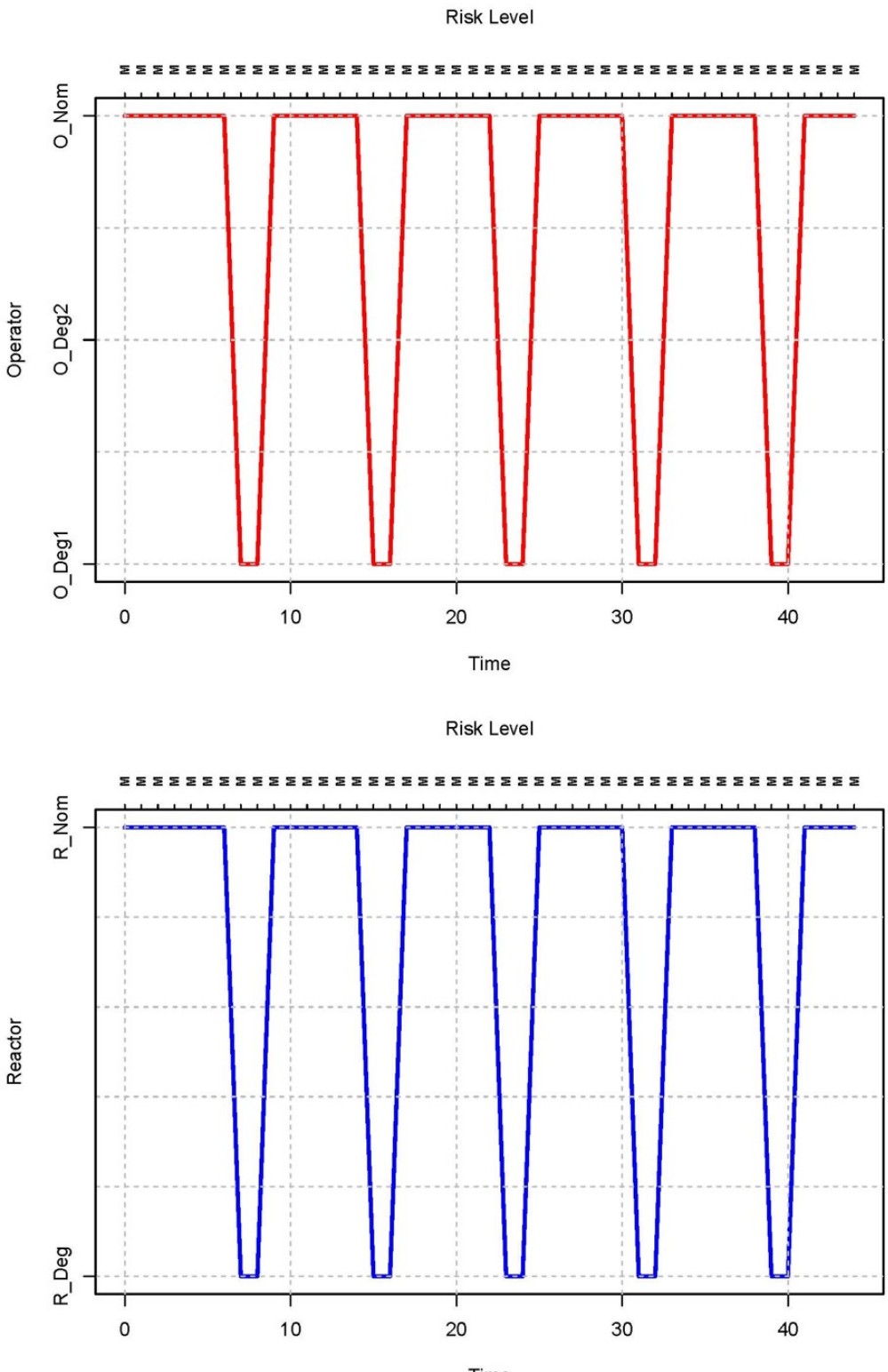

**Figure 17.** Agent's behavioral modes with the related Risk Level during simulation C3.

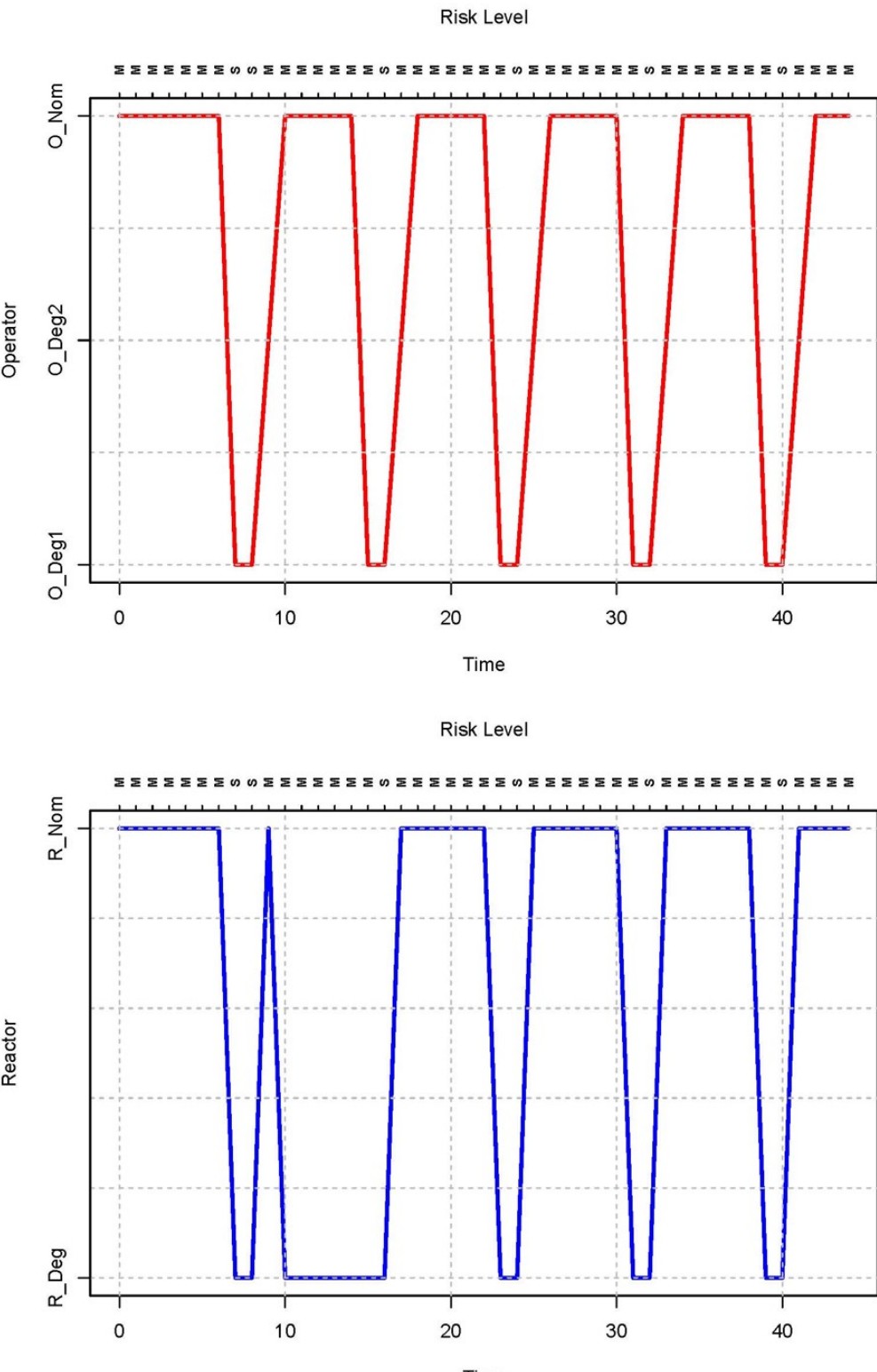

**Figure 18.** Agent's behavioral modes with the related Risk Level during simulation C4.

*8.2. Comparison with Other Modeling Approaches*

Numerical comparisons between the results obtained by current models are provided in [113]. Models studied in the comparison are system dynamics (SD) models, agent-based models (ABM), and discrete Event Simulation (DES) from a well-known case study (the spread of a disease).

The case study presented in this work (a reactor/operator system), is new. Therefore, similar studies based on alternative approaches are not available yet. The case study presented in this manuscript, which is a reactor/operator system, is considered novel. Therefore, similar studies based on alternative approaches are not available yet. The most relevant study found in [12] discusses the dynamic reliability of a steam generator using a Stochastic Hybrid Automaton with MCs, but this model does not provide full details about failure and behavioral modes, which are essential for dynamic risk analysis. Another comparison performed in [114] focuses on the limitations and capacities of different approaches, including Petri-nets and MCs to model a dynamic system rather than comparing the obtained numerical results. The work shows that the Petri-nets approach is a tool used for modeling systems with discrete events, but it is not adequate to be used for the continuous complex dynamic systems. This is demonstrated in the case of product level in the reactor agent. The strength of this method, such as modeling and simulating the system evolution by events occurrence, is not appropriate to be utilized in continuous dynamic systems. Numerically MCs generate the best values when used with stochastic events, but it requires a high computational effort in complex systems.

The same case study that was used in the paper was validated and verified when we simulated the classical ABM using Repast Simphony (2.0) open-source agent-based modeling and simulation platform, we got consistent results. A comparison between the proposed GDABM and the reference model (ABM) shows that GDABM provides higher levels of accuracy (Figure 19) (>15%). This is explained due to the detailed risk analysis performed in the proposed model, which reflects a better evaluation of the risk level with consideration of the temporal aspect (thanks to DFT).

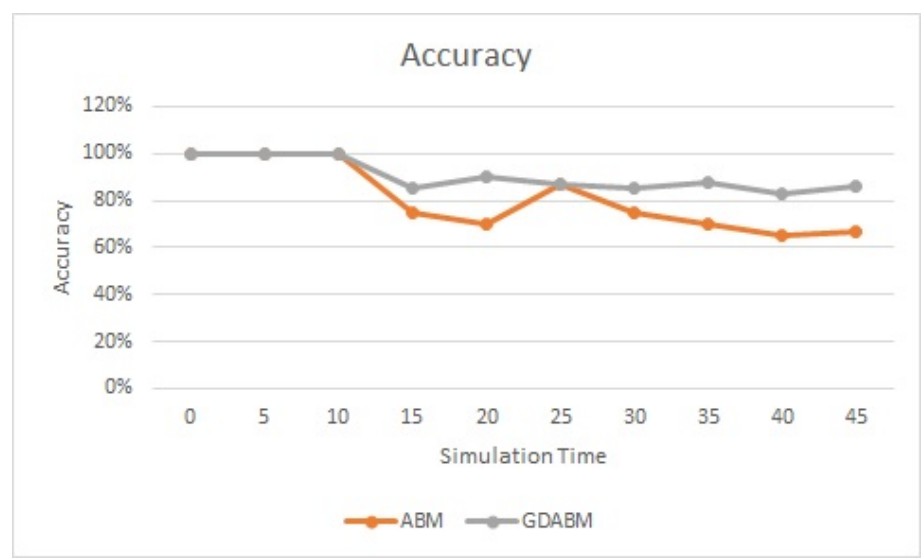

**Figure 19.** Comparison of the Accuracy for both ABM and GDABM.

Concerning the execution time (Figure 20), with the presence of the detailed analysis and the consideration of all analysis components, GDABM takes around 476 ms which is a bit slower than ABM (order of a few milliseconds) with consideration of a few number of agents (<10) that can raise up to 2 s for an important number of agents. This latency can be justified by the fact of representing the risk model in our proposed simulator and displaying more details about the active failure modes and their propagation in the system and the change in the informational state (attributes, behavioral mode, etc.) for every agent in the system.

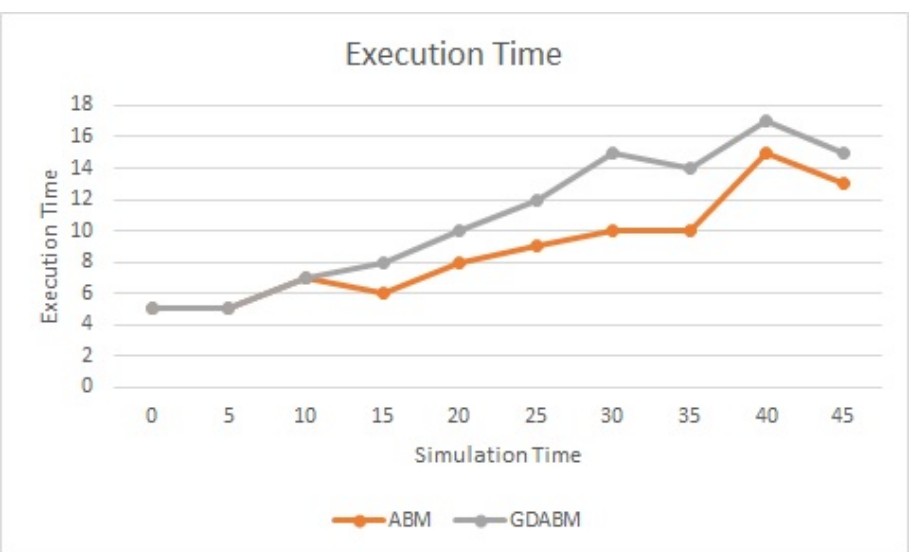

**Figure 20.** Comparison of the Execution times for both ABM and GDABM.

Regarding the response time (Figure 21), *GDABM* is measured to be faster than the reference model (ABM) in detecting and representing any change in the behavioral mode in real-time as it performs a real-time evaluation and analysis of possible failure modes. Furthermore, the details provided in the GDABM model allow the immediate detection of any failure, and, thus, the dynamic behavioral mode is up to date.

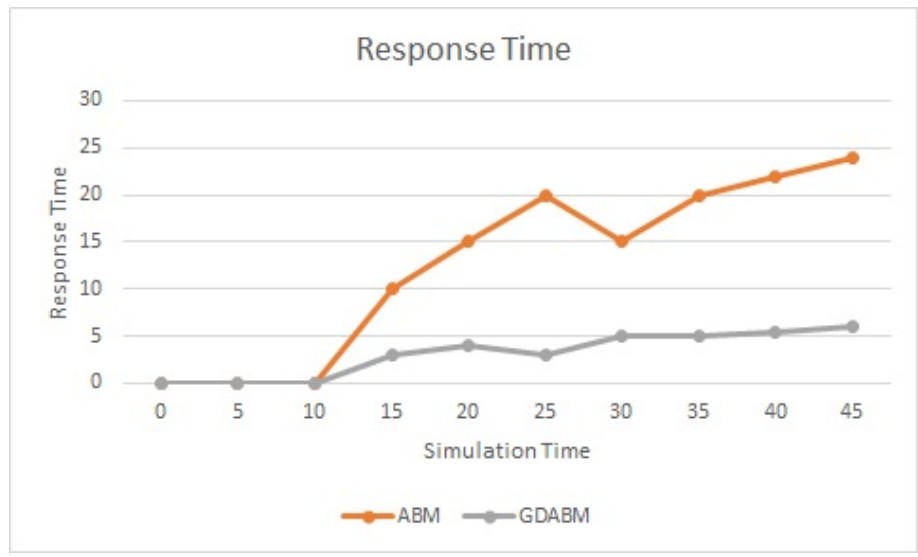

**Figure 21.** Comparison of the response times for both ABM and GDABM.

A common challenge was signed regarding the output whenever we increase the number of agents (hundreds) in the simulation and the interactions among them, a huge amount (MegaByte) of information and simulation results (change in the agent's behavioral mode, change in the agent's failure mode) should be extracted.

## 9. Conclusions

In this paper, a novel dynamic multi-agent model for risk analysis has been proposed and described thoroughly. The proposed model is called *Generic Dynamic Agent-Based Model* (GDABM) for risk analysis. This model represents the dynamic behavior of agents as a result of failure occurrence. It shows the failure propagation among the system's components as well as the failure dependencies between those components. Each agent in the modeled system has a set of activities, attributes, failure modes, and behavioral modes.

By studying the behavior of each agent in the system, GDABM was able to assess and analyze the risk of the entire system dynamically. GDABM thoroughly analyses dynamic systems in a coherent manner. It provides a graphical illustration of agents' behavioral modes with failure causes and outcomes and shows the direct relation between agents' behavioral modes transition and the activation/deactivation of a failure mode. A detailed case study of a chemical reactor/operator was provided. The case study used the GDABM to model various agents and their associated interactions. GDABM was able to simulate the behavior of the system in both nominal (failure-free) and degraded (failure) conditions. GDABM also analyzed the risk of the aforementioned systems. The goal of the proposed model is to analyze risks and to study the dynamic behavior of dynamic systems using multi-agents models. GDABM has proven to give very promising results when compared to the reference model (ABM) in terms of Accuracy (15%) and Response time (27%), for the execution time, GDABM signs an extra delay (13%) that can be accepted due to the real-time evaluation of active failure/behavioral modes.

The future work of this paper will be to use MCs for stochastic fuzzy failure, represent the population density in a dynamic way using a probability law, and test the model in different systems with a higher number of active agents and failure modes. Examples of such systems are dangerous good transportation, evacuation, and flood systems.

**Author Contributions:** Conceptualization, S.K.; methodology and approach, H.K.; software, S.K.; validation, H.K., S.K. and W.H.F.A.; formal analysis, S.K.; investigation, W.H.F.A.; resources, W.H.F.A.; data curation, S.K.; writing—original draft preparation, H.K.; writing—review and editing, W.H.F.A.; visualization, S.K.; supervision, S.K. All authors have read and agreed to the published version of the manuscript.

**Funding:** This research received no external funding.

**Conflicts of Interest:** The authors declare no conflict of interest.

## Abbreviations

The following abbreviations are used in this manuscript:

| | |
|---|---|
| ABM | Agent-Based models |
| GDABM | Generic Dynamic Agent-Based models |
| ABMS | Agent-Based Modeling and Simulation |
| SD | System Dynamics |
| DE | Discrete Events |
| PN | Petri Nets |
| MCs | Monte Carlo Simulation |
| RA | Risk Analysis |
| FTA | Fault tree Analysis |
| BM | Behavioral mode |
| FM | Failure mode |
| BFM | Boolean Failure mode |
| SFM | Stochastic Failure mode |
| CFM | Complex Failure mode |
| EFAC | External Failure Agent Communication |
| IFAC | Internal Failure Agent Communication |

# Appendix A

**Table A1.** Agents behavioral modes for C1 and C2.

| Configuration | Time Step | Reactor Behavioral Mode | Operator Behavioral Mode | Risk Level |
|---|---|---|---|---|
| C1 $V_{max} = 30$ $RR = 10,000$ | [0, 14] | $R_{Nom}$ | $O_{Nom}$ | M |
| | [14, 17] | $R_{Deg}$ | $O_{Deg1}$ | M |
| | [17, 18] | $R_{Nom}$ | $O_{Deg2}$ | S |
| | [18, 23] | $R_{Nom}$ | $O_{Nom}$ | M |
| | [23, 25] | $R_{Deg}$ | $O_{Deg1}$ | M |
| | [25, 45] | $R_{Nom}$ | $O_{Nom}$ | M |
| C2 $V_{max} = 30$ $RR = 15,000$ | [0, 14] | $R_{Nom}$ | $O_{Nom}$ | M |
| | [14, 15] | $R_{Deg}$ | $O_{Deg1}$ | M |
| | [15, 16] | $R_{Deg}$ | $O_{Deg1}$ | S |
| | [16, 17] | $R_{Deg}$ | $O_{Deg1}$ | S |
| | [16, 17] | $R_{Deg}$ | $O_{Deg2}$ | H |
| | [17, 18] | $R_{Nom}$ | $O_{Deg2}$ | S |
| | [18, 19] | $R_{Nom}$ | $O_{Deg2}$ | S |
| | [19, 22] | $R_{Nom}$ | $O_{Deg2}$ | S |
| | [22, 23] | $R_{Nom}$ | $O_{Deg2}$ | M |
| | [23, 24] | $R_{Deg}$ | $O_{Deg2}$ | M |
| | [24, 25] | $R_{Deg}$ | $O_{Deg2}$ | S |
| | [25, 31] | $R_{Nom}$ | $O_{Deg2}$ | M |
| | [31, 32] | $R_{Deg}$ | $O_{Deg2}$ | M |
| | [32, 33] | $R_{Deg}$ | $O_{Deg2}$ | S |
| | [33, 34] | $R_{Deg}$ | $O_{Deg2}$ | H |
| | [34, 35] | $R_{Nom}$ | $O_{Deg2}$ | S |
| | [35, 40] | $R_{Nom}$ | $O_{Deg2}$ | M |
| | [40, 41] | $R_{Deg}$ | $O_{Deg2}$ | M |
| | [41, 42] | $R_{Deg}$ | $O_{Deg2}$ | S |
| | [42, 45] | $R_{Nom}$ | $O_{Deg2}$ | M |

**Table A2.** Agents behavioral modes for C3 and C4.

| Configuration | Time Step | Reactor Behavioral Mode | Operator Behavioral Mode | Risk Level |
|---|---|---|---|---|
| C3 $V_{max} = 20$ $RR = 10,000$ | [0, 7] | $R_{Nom}$ | $O_{Nom}$ | M |
| | [7, 9] | $R_{Deg}$ | $O_{Deg1}$ | M |
| | [9, 15] | $R_{Nom}$ | $O_{Nom}$ | M |
| | [15, 17] | $R_{Deg}$ | $O_{Deg1}$ | M |
| | [17, 23] | $R_{Nom}$ | $O_{Nom}$ | M |
| | [23, 25] | $R_{Deg}$ | $O_{Deg1}$ | M |
| | [25, 31] | $R_{Nom}$ | $O_{Nom}$ | M |
| | [31, 33] | $R_{Deg}$ | $O_{Deg1}$ | M |
| | [33, 39] | $R_{Nom}$ | $O_{Nom}$ | M |
| | [39, 41] | $R_{Deg}$ | $O_{Deg1}$ | M |
| | [41, 45] | $R_{Nom}$ | $O_{Nom}$ | M |

**Table A2.** *Cont.*

| Configuration | Time Step | Reactor Behavioral Mode | Operator Behavioral Mode | Risk Level |
|---|---|---|---|---|
| C4 | [0, 7] | $R_{Nom}$ | $O_{Nom}$ | M |
| $V_{max} = 20$ | [7, 9] | $R_{Deg}$ | $O_{Deg1}$ | S |
| $RR = 15,000$ | [9, 10] | $R_{Nom}$ | $O_{Deg2}$ | M |
| | [10, 15] | $R_{Nom}$ | $O_{Nom}$ | M |
| | [15, 16] | $R_{Deg}$ | $O_{Deg1}$ | M |
| | [16, 17] | $R_{Deg}$ | $O_{Deg1}$ | S |
| | [17, 18] | $R_{Nom}$ | $O_{Deg2}$ | M |
| | [18, 23] | $R_{Nom}$ | $O_{Nom}$ | M |
| | [23, 24] | $R_{Deg}$ | $O_{Deg1}$ | M |
| | [24, 25] | $R_{Deg}$ | $O_{Deg1}$ | S |
| | [25, 26] | $R_{Nom}$ | $O_{Deg2}$ | M |
| | [26, 31] | $R_{Nom}$ | $O_{Nom}$ | M |
| | [31, 32] | $R_{Deg}$ | $O_{Deg1}$ | M |
| | [32, 33] | $R_{Deg}$ | $O_{Deg1}$ | S |
| | [33, 34] | $R_{Nom}$ | $O_{Deg2}$ | M |
| | [34, 39] | $R_{Nom}$ | $O_{Nom}$ | M |
| | [39, 40] | $R_{Deg}$ | $O_{Deg1}$ | M |
| | [40, 41] | $R_{Deg}$ | $O_{Deg1}$ | S |
| | [41, 42] | $R_{Nom}$ | $O_{Deg2}$ | M |
| | [42, 45] | $R_{Nom}$ | $O_{Nom}$ | M |

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
