# Peer review of "A Novel Dynamic Approach for Risk Analysis and Simulation Using Multi-Agents Model"

_applsci, doi:10.3390/app12105062_

Round 1

Reviewer 1 Report

The article is devoted to developing a dynamic approach to modeling stochastic systems using dynamic fault trees. The study's relevance is that the methods of static risk analysis are not suitable for modeling dynamic stochastic systems since they are not based on time. Therefore, the authors propose a new dynamic approach to modeling such stochastic systems using dynamic fault trees in the article. A Generic Dynamic Agent-Based Model based on the agent-based approach is proposed. The Generic Dynamic Agent-Based Model can model agents of a dynamic system both in nominal (failsafe) and degraded (failure) modes. The Generic Dynamic Agent-Based Model shows failure propagation between system elements and provides complete information about system configurations. The experimental study presented in the article shows the possibilities of the Generic Dynamic Agent-Based Model for modeling and studying risk analysis for such dynamic systems. In this example, the Generic Dynamic Agent-Based Model models a risk analysis for a chemical reactor/operator. Detailed agent behaviors and failure modes are provided with different scenarios, including different timestamps. The Generic Dynamic Agent-Based Model was able to periodically study the dynamics of the operator and reactor operating modes based on active failure modes.

Despite the satisfactory quality of the article, some shortcomings need to be corrected.

  1. The abstract should be rewritten. The numerical results obtained by the authors should be added. Some information in the abstract now is repeated several times.
  2. It is recommended to include the logical-combinatorial approach to the current research analysis because it could be effectively used in solving the proposed task.
  3. The state-of-the-art methods, models, and approaches should be separated from the ones proposed by the authors.
  4. The data used for the experimental study should be described in more detail.
  5. Tables 8-10 can be put in supplementary materials. Authors can leave only interpretations of the experiments.
  6. The discussions section should include the comparative study of obtained results with other research and approaches.
  7. The conclusion section should contain numerical results obtained by the authors.
  8. Authors do not consider one of the valuable features of agent-based simulation: knowledge bases and the ability to make decisions. It is recommended to describe that in the current research analysis section, e.g. doi: 10.1109/ELIT.2019.8892307

In summarizing my comments, I recommend that the manuscript is accepted after major revision. 

Reviewer 2 Report

Major Revisions

  • The introduction is short, no antecedents of similar studies are established, and no contrast is made between the current methodologies and what the authors propose, so it must be rewritten.

  • There are some concepts from sections 2.2 and 2.3 that are well known and can be summarized

  • The authors should carry out a better and more careful discussion of the results obtained
  • I consider that the assumptions established for the case study are not explicit. How is it that the information was obtained? The authors must establish under what premises they have obtained the information

  • A crucial aspect is that when performing an analysis through Turnitin, the manuscript registered a 21% similarity, so the authors must modify this aspect of their document.

  • I consider that your manuscript's length is too long because it defines concepts or ideas that are well known, so I suggest the author reduce the concepts and focus on further developing the methodology and the results stage.

Minor Revisions:

  • Authors must review the writing and grammar of the entire manuscript.

  • The quality of the figures should be improved

Author Response

Kindly check the attached document

Round 2

Reviewer 1 Report

Thanks for the authors for their grounded work. All the comments and recommendations were considered. In my opinion, now the paper can be accepted.

Reviewer 2 Report

After reviewing the manuscript and the response letter from the authors where they explain each of the questions and suggestions made, I can issue a favorable recommendation for the publication of this manuscript.

This manuscript is a resubmission of an earlier submission. The following is a list of the peer review reports and author responses from that submission.